# Polycyclic Polyprenylated Acylphloroglucinol Derivatives from *Hypericum acmosepalum*

**DOI:** 10.3390/molecules24010050

**Published:** 2018-12-23

**Authors:** Jiao Wang, Mengjiao Shi, Jiajia Wang, Jin Li, Tengfei Ji

**Affiliations:** 1The Key Laboratory of Plant Stress Biology in Arid Land, College of Life Sciences, Xinjiang Normal University, Urumqi 830054, China; 17364768898@163.com; 2State Key Laboratory of Bioactive Substance and Function of Natural Medicines, Institute of Materia Medica, Chinese Academy of Medical Sciences & Peking Union Medical College, Beijing 100050, China; 18363030786@163.com (M.S.); wangjiajia@imm.ac.cn (J.W.)

**Keywords:** *Hypericum acmosepalum*, polycyclic polyprenylated acylphloroglucinols (PPAPs), neuroprotective effect

## Abstract

*Hypericum acmosepalum* belongs to the *Hypericum* genus of the Guttiferae family. The characteristic components in *Hypericum* are mainly a series of polycyclic polyprenylated acylphloroglucinols (PPAPs), flavonoids, and xanthones. Among them, the PPAPs have received much attention due to their novel structures and diverse pharmacological activities and have become hot spots in organic chemistry and medicinal chemistry. However, there are few reports about the chemical constituents of *Hypericum acmosepalum* at present, especially the PPAPs. This research is dedicated to the study of the air-dried aerial parts of *Hypericum acmosepalum*, which were extracted with 95% EtOH under reflux, then suspended and successively partitioned with petroleum ether and ethyl acetate. Five PPAP derivatives were obtained using various chromatographic techniques, and their structures were determined by NMR spectroscopic data, including two new phloroglucinol derivatives, hyperacmosin A (**1**) and hyperacmosin B (**2**). Those compounds were evaluated for their neuroprotective effect using two models.

## 1. Introduction 

Plants of the genus *Hypericum* of the family Guttiferae, which includes 484 species, mostly herbs or shrubs, are widely distributed in various regions of the world. There are 55 species and eight subspecies in China, and there are 27 species and four subspecies in Yunnan Province alone [1,2,3]. *Hypericum* plants have diverse pharmacological applications as folk medicines due to the diverse chemical constituents [4,5,6]. As a classical natural product from this family, polycyclic polyprenylated acylphloroglucinols (PPAPs) have intriguing chemical structures [7,8,9] and various biological activities, such as anti-neurodegenerative, anti-HIV, tumor inhibitory, antioxidative, antidepressant, and so on [10,11,12,13,14,15]. 

*Hypericum acmosepalum* N. Robson, commonly known as Xiangzhen tree in Yunnan Province, belongs to the *Hypericum* genus [16]. It mostly grows in Yunnan, Guizhou, and Sichuan Province, and is used in folk medicine to treat inflammation and hepatitis [17]. Currently, there are few studies on the chemical constituents of *H. acmosepalum* [18]. As a part of an ongoing research program aimed at the isolation, structural characterization, and pharmacological evaluation of bioactive PPAPs from the *Hypericum* genus [6,19,20], a preliminary study was carried out on the petroleum ether-soluble part of the 95% EtOH extract from the air-dried aerial parts of *H. acmosepalum*, and obtained five PPAP derivatives. The neuroprotective effect of those compounds was evaluated in human neuroblastoma SK-N-SH cells. 

## 2. Results and Discussion

The 95% EtOH extract of the dried aerial parts of *H. acmosepalum* was partitioned successively with petroleum ether and ethyl acetate (EtOAc). The petroleum ether soluble part was subjected to silica gel column chromatography to give two new compounds (**1**–**2**) and three known compounds: Oxepahyperforin (**3**) [21], sampsonione O (**4**) [22], sampsonione L (**5**) [23,24] (Figure 1, ^1^H- and ^13^C-NMR data for **3**–**5** are found in the Appendix A). The structure of the new compounds were elucidated by the analysis of spectroscopic data and the known compounds were concluded by comparison of the NMR data with the published literature. 

Compound **1** was acquired as a colorless oil. The molecular formula of **1** was determined as C_35_H_52_O_5_ on the basis of HRESIMS (high resolution electrospray ionization mass spectrum) data (*m/z* 553.3883 [M + H]^+^, calculated 553.3888, see Appendix A), with ten degrees of unsaturation. Its IR (infrared spectrum, see Appendix A) absorptions implied the presence of hydroxy (3440 cm^−1^) and carbonyl groups (1725, 1642, and 1618 cm^−1^). The ^1^H-NMR spectroscopic data (Table 1) of hyperacmosin A showed the presence of three olefinic protons signals of isopentenyl (*δ*_H_ 4.83 (1H, t, *J* = 7.0 Hz), 5.04 (1H, s), 5.04 (1H, s)), two doublet methyl signals of the isopropyl group (*δ*_H_ 1.16 (3H, d, *J* = 6.5 Hz), 1.03 (3H, d, *J* = 6.5 Hz)), nine singlet methyl signals. Additionally, the data showed characteristic signals of furan-type hyperforin compounds: An oxygen-bearing methine signal of furan nucleus (*δ*_H_ 4.62 (1H, dd, *J* = 11.5, 10.0 Hz)). The ^13^C-NMR data (Table 1) showed 35 carbon signals, which were similar to those of furohyperforin [19] except for the position of C-(14–18): The fragment C-(14–18) of furohyperforin was located at C-34, but the C-(14–18) of **1** was located at C-28. Three carbonyl carbons (*δ*_C_ 208.6, 206.8, 191.5), six olefinic carbons (*δ*_C_ 138.0, 132.9, 131.1, 124.5, 124.4, 119.5), the characteristic carbon signals of *α*, *β*-unsaturated carbonyl group (*δ*_C_ 191.5, 172.7, 119.5), an oxygen-bearing methine (*δ*_C_ 93.5) and three quaternary carbon signals on the skeleton (*δ*_C_ 72.0, 62.1, 46.5). These data indicated that **1** was a furan-type hyperforin derivative. In the HMBC (heteronuclear multiple bond correlation) and HSQC (heteronuclear single quantum correlation) spectra (Figure 2), based on two classic methyl signals in a double peak of the isopropyl group in ^1^H-NMR, the chemical shift of C-12 and C-13 were determined. H-11 was correlated with C-10, C-12, C-13; H-12 was correlated with C-10, C-11, C-13; H-13 was correlated with C-10, C-11, C-12; the fragment C-(10–13) was determined. Based on olefinic proton H-30, the chemical shift of C-30 was determined, H-30 was correlated with C-29, C-31, C-32; H-32 was correlated with C-30, C-31, C-33; H-33 was correlated with C-30, C-31, C-32, in addition, H-29 was correlated with C-6, C-7, C-8, C-30 and C-31, suggesting that the branch of C-(29–33) was connected to C-7. The correlations of H-34 with C-1, C-7, C-8, C-35 and H-35 with C-1, C-7, C-8, C-34 showed that two methyl groups were situated at C-8. Additionally, the correlations of H-22 with C-20, C-21, C-23; H-23 with C-20, C-21, C-22; H-20 with C-19, C-22, C-23; H-19 with C-4, C-20, C-21 determined the fragment of C-(19-23). The correlations of H-17 with C-15, C-16, C-18; H-18 with C-15, C-16, C-17; H-15 with C-14, C-18; H-14 with C-15, C-26 determined the fragment of C-(14-18); and the correlations of H-27 with C-25, C-26, C-28; H-28 with C-14, C-15, C-(25-27); H-25 with C-24, C-27, C-28; H-24 with C-(2–4), C-25, C-26 determined the fragment of C-(24-28), which showed that geranyl alkenyl was situated at C-3. In the ROESY (Rotating frame Overhauser Enhancement Spectroscopy) spectrum (see Appendix A) of **1**, significant correlations were observed between H-20 (*δ*_H_ 4.62), H-13 (*δ*_H_ 1.03), H-19 (*δ*_H_ 2.88), H-22 (*δ*_H_ 1.36) and H-23 (*δ*_H_ 1.27); H-7 (*δ*_H_ 1.97), H-22 (*δ*_H_ 1.36), H-30 (*δ*_H_ 5.04) and H-34 (*δ*_H_ 1.38). Hence, the relative configuration was determined as shown in Figure 2. To assign the absolute configuration of **1**, the electronic circular dichroism (ECD) experiment was conducted by the time-dependent density functional theory (TDDFT) method [20,25]. Based on a large number of previous studies on PPAPs in our laboratory, the chirality of C-1, 5, 7 in the skeleton structure of **1** had been assigned to be (1R, 5S, 7S) by comparing the ECD spectra (Figure 3) to that of hyperscabrones A [9]. In the ROESY spectrum of **1**, the correlations between H-20 (*δ*_H_ 4.62) and H-13 (*δ*_H_ 1.03) indicated that the orientation of the H-20 was *β*-oriented. Therefore, the absolute configuration of **1** was determined to be (1R, 5S, 7S, 20R) as shown in Figure 4, and was named hyperacmosin A.

Compound **2** was obtained as a colorless oil. The molecular formula of **2** was determined as C_35_H_52_O_5_ on the basis of HRESIMS data (*m/z* 553.3892 [M + H]^+^, calculated 553.3888, see Appendix A), with ten degrees of unsaturation. Its IR (see Appendix A) absorptions implied the presence of hydroxy (3467 cm^−1^) and carbonyl groups (1726 and 1618 cm^−1^). The ^1^H-NMR and ^13^C-NMR data of **2** were similar to those of **1**, except for H-15 and 30 (**2**: *δ*_H_ 5.05 and 4.88, **1**: *δ*_H_ 4.83 and 5.04), 22 and 34 (**2**: *δ*_H_ 1.38 and 1.36, **1**: *δ*_H_ 1.36 and 1.38), 23 and 35 (**2**: *δ*_H_ 1.25 and 1.28, **1**: *δ*_H_ 1.27 and 1.25). In the HMBC spectra, H-11 was correlated with C-10, C-12, C-13; H-32 was correlated with C-7, C-30, C-31, C-33; H-29 was correlated with C-6, C-8, C-30 and C-31. In addition, the correlations of H-34 with C-1, C-7, C-35; H-20 with C-3, C-4, C-19, C-21, C-22, C-23; H-19 with C-2, C-3, C-4, C-8, C-10, C-20, C-21. The correlations of H-15 with C-17, C-27, C-28; H-14 with C-15, C-16, C-26; and the correlations of H-28 with C-14, C-15, C-(25-27); H-24 with C-2, C-5, C-6, C-25, C-26. The ROESY spectra (see Appendix A) revealed the correlations (Figure 5) between H-20 (*δ*_H_ 4.80), H-19 (*δ*_H_ 2.94), H-22 (*δ*_H_ 1.38), H-23 (*δ*_H_ 1.25) and H-29 (*δ*_H_ 2.21); H-7 (*δ*_H_ 1.96), H-13 (*δ*_H_ 1.15), H-22 (*δ*_H_ 1.38), H-27 (*δ*_H_ 1.67) and H-35 (*δ*_H_ 1.28). Thus, the relative configuration was determined as shown in Figure 5. In the ECD spectra, the chirality of C-1, 5, 7 in the skeleton structure of **2** was consistent with compound **1** because the spectra of **2** had a high degree of similarity to the spectra of **1** (Figure 6). Furthermore, the ROESY correlation between H-20 (*δ*_H_ 4.62) and H-29 (*δ*_H_ 2.21) indicated that the orientation of the H-20 was *α*-oriented. Therefore, the absolute configuration of **2** was determined to be (1R, 5S, 7S, 20S), and was named hyperacmosin B.

The five compounds were evaluated for neuroprotective effect in two models: l-glutamic acid (L-Glu) and oxygen-glucose deprivation (OGD) by the MTT method [26]. Donepezil and PHPB (potassium 2-(1-hydroxypentyl)-benzoate) were used as the positive controls. l-glutamic acid and sodium hyposulfite were used as the damage agents. In the model of l-Glu, the cell survival rate of the positive control groups had increased compared with the control group. In experiment groups, hyperacmosin A (**1**) and sampsonione O (**4**) increased the survival rates of the SK-N-SH cells to 68.20%, 80.90% compared with the control group respectively (Table 2), which showed remarkable neuroprotection activity. In the model of OGD, PHPB increased the survival rates of the SK-N-SH cells to 80.30% compared with the control group (Table 2). While the cell survival rate of hyperacmosin B, oxepahyperforin and sampsonione L had increased, those compounds did not exhibit obvious neuroprotective activity against OGD induced-injury on SK-N-SH cells compared with the control group. 

## 3. Materials and Methods 

### 3.1. General Information

Optical rotations were measured on a JASCO P-2000 polarimeter (Jasco Inc., Tokyo, Japan) using methanol as a solvent. UV spectra were determined with a JASCO V-650 spectrophotometer (Jasco Inc., Tokyo, Japan). IR spectra were recorded on a Nicolet 5700 FT-IR spectrometer (Thermo Fisher Scientific, Waltham, MA, USA). ECD and NMR spectra were obtained on Varian Inova-500 spectrometer (Varian Inc., Palo Alto, CA, USA) and a Mecury-400 spectrometer (Varian Inc., Palo Alto, CA, USA) with trimethylsilane (TMS) as an internal standard and CDCl_3_ as a solvent. HRESIMS spectra were performed on an Agilent 1100 LC/MSD Trap SL mass spectrometer (Agilent Technologies Ltd., Santa Clara, CA, USA). Preparative HPLC experiments were carried out on a preparative YMC-Pack ODS-A column (250 × 20 mm, 5 μm, YMC, Kyoto, Japan). Silica gel (100–200 mesh and 200–300 mesh, Qingdao Haiyang Chemistry Company, Qingdao, China), silica gel H (Qingdao Haiyang Chemistry Company, Qingdao, China), MCI gel CHP20P (35−75 μm, Mitsubishi Chemical Corp, Japan), ODS (50 μm, YMC, Kyoto, Japan), and Sephadex LH-20 (40−70 μm, Healthcare Bio-Sciences AB, Uppsala, Sweden) were used for column chromatography and silica gel GF-254 (Qingdao Haiyang Chemistry Company, Qingdao, China) was used for TLC. 3-(4,5-dimethylthiazol-2-yl)-2,5-diphenyltetrazolium bromide (MTT) were purchased from Sigma-Aldrich (Saint Louis, MO, USA). Other chemicals used in the study were of analytical grade (Beijing Chemical works, Beijing, China). Fetal bovine serum (FBS) was obtained from Invitrogen Corporation (Carlsbad, CA, USA). Penicillin and streptomycin were purchased from Sigma (St. Louis, MO, USA).

### 3.2. Plant Material

The air-dried aerial parts of *H. acmosepalum* were collected from the Wenshan, Yunnan Province, China, in July 2016. The plant was identified by Prof. Lin Ma. A voucher specimen (No. ID-S-2764) was deposited in the herbarium of the Institute of Materia Medica, Chinese Academy of Medical Sciences.

### 3.3. Extraction and Isolation

Air-dried and powdered aerial parts of *H. acmosepalum* (17.0 kg) were extracted three times, with 95% EtOH (170 L per extraction) under reflux. The 95% EtOH extracts were concentrated in vacuo to provide a crude extract (1.8 kg), which was suspended in water and then partitioned with petroleum ether and EtOAc successively. The petroleum ether soluble part (500.0 g) was separated by silica gel column chromatography (200−300 mesh) in reduced pressure and eluted with a gradient of petroleum ether–EtOAc (1:0 to 0:1) to obtain 11 fractions (Fr. 1–Fr. 11).

Fr. 1 (254.0 g) was subjected to column chromatography over MCI gel in reduced pressure, eluting with EtOH-H_2_O (75% to 95%), to give 6 fractions (Fr. 1A–Fr. 1F). Fr. 1C (94.0 g) was loaded onto silica gel column chromatography, eluting with petroleum ether–EtOAc (1:0 to 0:1), to give 20 fractions (Fr. A–Fr. t), Fr. m (5.7 g) was further separated by silica gel H column chromatography and eluted with petroleum ether-CH_2_Cl_2_ (9:1) and petroleum ether-Et_2_O (20:1) repeatedly to give compound **2** (17.0 mg). Fr. r (9.1 g) was further separated by silica gel H column chromatography and eluted with petroleum ether-EtOAc (1:0 to 0:1) and petroleum ether-Et_2_O (20:1) to yield 14 fractions (Fr. r1–Fr. r14). Compound **1** (9.4 mg) was gained from both Fr. r5 and Fr. r6 (2.4 g) by ODS column chromatography (MeOH-H_2_O), silica gel H column chromatography (petroleum ether–EtOAc) and semipreparative HPLC.

Fr. 3–5 (104.0 g) were subjected to column chromatography over MCI gel in reduced pressure, eluting with EtOH–H_2_O (70% to 95%), to give 12 fractions (Fr. A–Fr. L). Fr. A (22.0 g) was loaded onto MPLC (octa decylsilyl silicion, ODS) column chromatography eluted by MeOH-H_2_O (70% to 100%), which gave 7 fractions (Fr. A1–Fr. A7). Fr. A4 (7.5 g) and Fr. A5 (6.0 g) were further separated by silica gel H column chromatography and eluted with petroleum ether-EtOAc (1:0 to 0:1) respectively to give several subfractions. Each subfraction of Fr. A4 was further purified by silica gel H column chromatography with petroleum ether-EtOAc (9:1), by HPLC (RP C18, 85% MeOH) and by Sephadex LH-20 (MeOH) to give compound **4** (20.0 mg) and compound **5** (30.0 mg). Each subfraction of Fr. A5 was further purified by Sephadex LH-20 (MeOH), silica gel H column chromatography with petroleum ether-EtOAc (9:1), and recrystallization to give compound **3** (27.0 mg).

### 3.4. Spectroscopic Data

Hyperacmosin A (1): Colorless oil, [*α*]^20^_D_ = +87.8 (*c* = 0.337, MeOH). UV: (MeOH) *λ*_max_ (log *ε*) 209.8 (1.96), 280.0 (2.64) nm. IR (FT-IR Microscope Transmission method): *υ*_max_ 3440, 3229, 2973, 2927, 1726, 1642, 1619, 1444, 1405, 1249, 1149, 1092, 969, 846, 802 cm^−1^. ^1^H-NMR (500 MHz, CDCl_3_) and ^13^C-NMR (125 MHz, CDCl_3_) data are shown in Table 1. HRESIMS (MeOH): 553.3883 ([M + H]^+^, C_35_H_52_O_5_; calculated 553.3888). HSQC, HMBC and ROESY (500 MHz, CDCl_3_) spectroscopic data can be found in the Appendix A. ECD (MeOH) *λ*(Δ *ε*) 209 (+9.14), 275.5 (−23.64), 307.5 (+14.68) nm.

Hyperacmosin B (2): Colorless oil, [*α*]^20^_D_ = −53.0 (*c* = 0.502, MeOH). UV: (MeOH) *λ*_max_ (log *ε*) 203.4 (0.57), 282.2 (0.28) nm. IR (FT-IR Microscope Transmission method): *υ*_max_ 3467, 2971, 2926, 1727, 1619, 1447, 1381, 1240, 1092, 966, 842 cm^−1^. ^1^H-NMR (400 MHz, CDCl_3_) and ^13^C-NMR (125 MHz, CDCl_3_) data are shown in Table 1. HRESIMS (MeOH): 553.3892 ([M + H]^+^, C_35_H_52_O_5_; calculated 553.3888). HSQC, HMBC and ROESY (500 MHz, CDCl_3_) spectroscopic data can be found in the Appendix A. ECD (MeOH) *λ*(Δ *ε*) 209 (+8.03), 278 (−27.48), 310 (+12.08) nm.

### 3.5. Neuroprotection Bioassays

The neuroprotection of compounds was determined by the MTT method in SK-N-SH cells, grown in Dulbeco’s modified containing 10% FBS, penicillin (100 U/mL) and streptomycin (100 μg/mL). Cell cultures were incubated at 37 °C under a 5% CO_2_ atmosphere. The compounds and positive control were dissolved with DMSO (10 μmol/L) respectively. The cells were seeded onto 96-well microplates at a density of 1 × 10^5^ cells/well. After incubation at 37 °C for 24 h under a 5% CO_2_ atmosphere, the cells were pre-incubated with compounds and positive controls respectively for 1 h. In the l-Glu, the DMEM (Dulbecco’s modified Eagle’s medium) without l-Glu was added to the cells of the normal group, the DMEM containing l-Glu (the final concentration of 27 mM) was added to the cells of other groups. After 4 h of co-incubation, 100 μL of MTT (0.5 mg/mL) were added to each well after the withdrawal of the culture medium and were incubated for an additional 4 h at 37 °C. The resulting formazan crystals were dissolved in 150 μL of DMSO after aspiration of the culture medium. The optical density (OD) of the formazan solution was measured on a microplate reader at 570 nm. In the OGD, the l-DMEM (low-sugar Dulbecco’s modified Eagle’s medium) without sodium hyposulfite was added to the cells of the normal group, the l-DMEM containing sodium hyposulfite (the final concentration of 3.5 mM) was added to the cells of other groups, and then incubated for 24 h. After 24 h, MTT solution (0.5 mg/mL) was added for 4 h at 37 °C. Finally, the formazan crystals were solubilized by DMSO and were spectrophotometrically measured at 570 nm. All data presented in our study were obtained from three independent experiments. Survival rate (%) was obtained by the following formula: Improved survival rate (%) = (survival rate of the experimental group-survival rate of the control group)/Survival rate of the control group.

## 4. Conclusions

In this study, five phloroglucinols were isolated from the petroleum ether soluble part of the 95% EtOH extract from *H. acmosepalum*, including two new phloroglucinols and three known compounds. In the neuroprotection screening, the compound **1** (hyperacmosin A) and compound **4** (sampsonione O) showed significant neuroprotective activity in L-Glu induced-injury on human neuroblastoma SK-N-SH cells. However, those compounds did not exhibit neuroprotective activity in the OGD. 

In summary, PPAPs research is currently in focus, but the low polarity and instability of the PPAPs greatly increase the difficulty in their separation and structure determination. Additionally, this work provides preliminary evidence for the neuroprotective function of these compounds, but further investigations of this neuroprotective function and other activities are warranted. By elucidating how these molecules interact, this work provides the necessary data to explore the relationship between neuroprotective activity and molecular structure. 

## Figures and Tables

**Figure 1 molecules-24-00050-f001:**
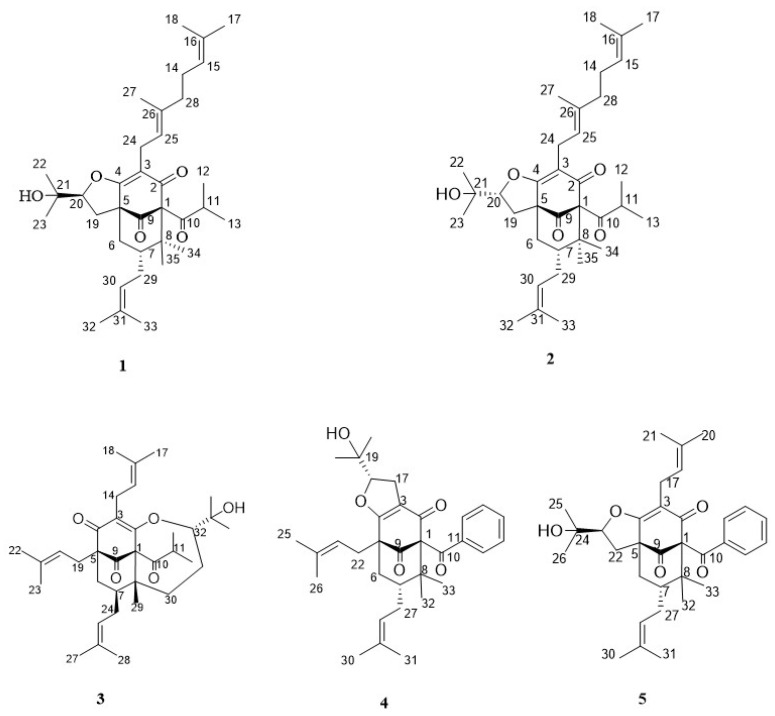
Structure of compounds **1**–**5**.

**Figure 2 molecules-24-00050-f002:**
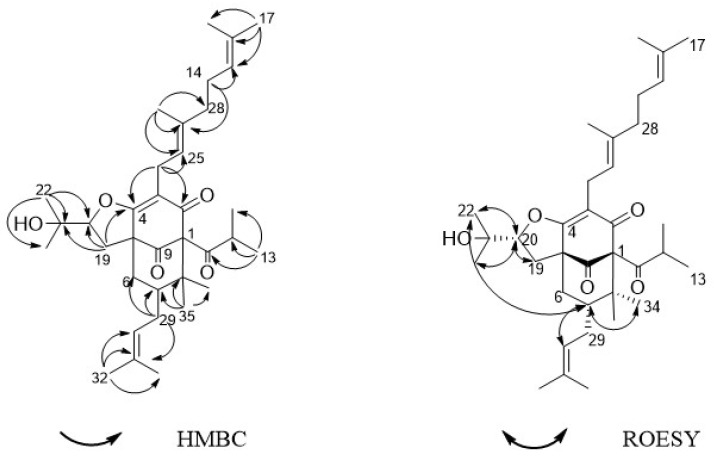
Selected key HMBC and ROESY correlations for **1**.

**Figure 3 molecules-24-00050-f003:**
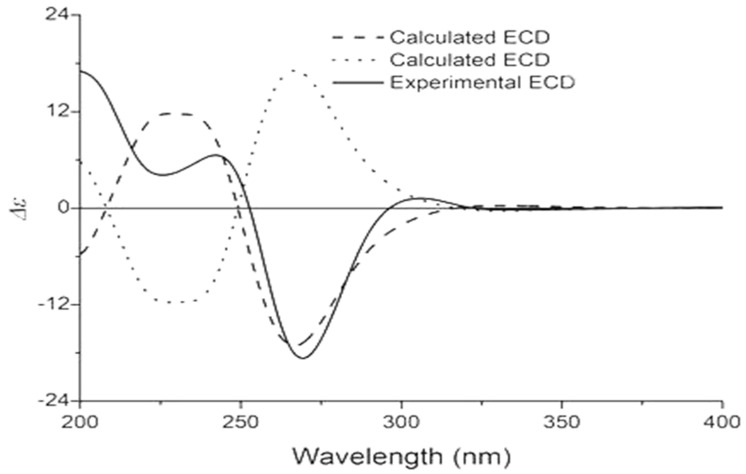
Calculated and experimental ECD spectra of hyperscabrones A in methanol [9].

**Figure 4 molecules-24-00050-f004:**
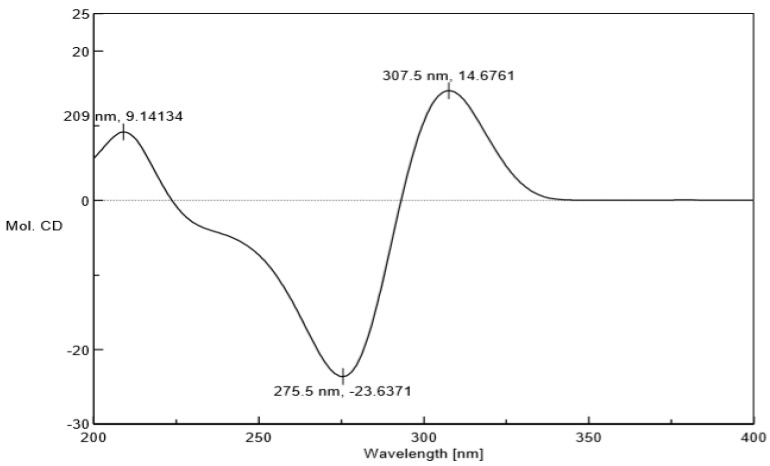
Experimental ECD spectra of **1** in methanol.

**Figure 5 molecules-24-00050-f005:**
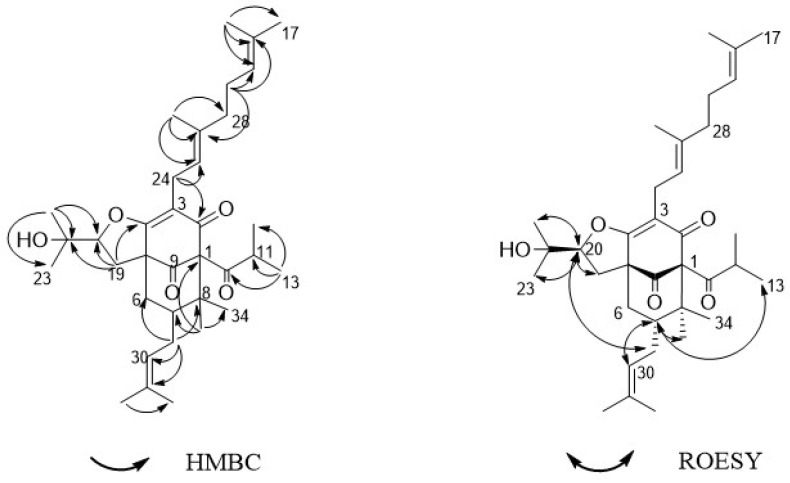
Selected key HMBC and ROESY correlations of **2**.

**Figure 6 molecules-24-00050-f006:**
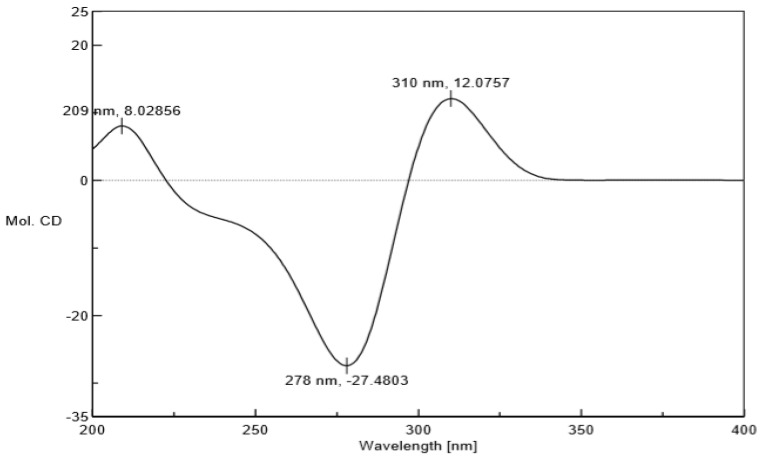
Experimental ECD spectra of **2** in methanol.

**Table 1 molecules-24-00050-t001:** ^1^H- and ^13^C-NMR data for **1** and **2** in CDCl_3_.

Position	Type	1		2	
		*δ*_C_ *^a^*	*δ*_H_ (*J* in Hz)* ^b^*	*δ*_C_ *^a^*	*δ*_H_ (*J* in Hz)* ^c^*
1	C	72.0	-	71.7	-
2	C	191.5	-	191.2	-
3	C	119.5	-	120.1	-
4	C	172.7	-	173.1	-
5	C	62.1	-	62.3	-
6	CH_2_	38.6	2.05, m 1.97, m	38.7	2.13, m 1.96, m
7	CH	48.1	1.97, m	48.2	1.96, m
8	C	46.5	-	46.9	-
9	C	206.8	-	206.5	-
10	C	208.6	-	208.6	-
11	CH	39.9	2.39, m	39.8	2.49, m
12	CH_3_	20.9	1.16, d (6.5)	21.1	1.11, d (5.2)
13	CH_3_	20.6	1.03, d (6.5)	21.0	1.15, d (5.2)
14	CH_2_	26.7	2.03, m	26.7	2.01, m
15	CH	124.4	4.83, t (7.0)	124.3	5.05, m
16	C	131.1	-	131.2	-
17	CH_3_	25.7	1.64, s	25.7	1.64, s
18	CH_3_	17.6	1.57, s	17.6	1.57, s
19	CH_2_	27.6	2.88, m	27.2	2.94, m
20	CH	93.5	4.62, dd (11.5, 10.0)	93.9	4.80, t (8.0)
21	C	70.7	-	71.2	-
22	CH_3_	26.5	1.36, s	26.7	1.38, s
23	CH_3_	25.1	1.27, s	25.3	1.25, s
24	CH_2_	29.2	2.46, m	29.3	2.49, m
25	CH	119.3	5.04, s	119.3	5.05, m
26	C	138.0	-	138.0	-
27	CH_3_	16.4	1.68, s	16.3	1.67, s
28	CH_2_	40.0	1.97, m	40.0	1.96, m
29	CH_2_	29.6	2.17, m 1.80, m	29.5	2.21, m 1.96, m
30	CH	124.5	5.04, s	124.5	4.88, t (5.2)
31	C	132.9	-	132.7	-
32	CH_3_	25.8	1.64, s	25.8	1.66, s
33	CH_3_	17.9	1.52, s	17.9	1.55, s
34	CH_3_	23.7	1.38, s	23.2	1.36, s
35	CH_3_	26.3	1.25, s	26.7	1.28, s

*^a^* Recorded at 125 MHz; *^b^* Recorded at 500 MHz; *^c^* Recorded at 400 MHz.

**Table 2 molecules-24-00050-t002:** Neuroprotection activity of **1** to **5** on human neuroblastoma SK-N-SH cells.

Compounds	L-Glu Model			Compounds	OGD Model		
	Mean ± SD (*n* = 3)	Survival Rate (%)	Improved Survival Rate (%)		Mean ± SD (*n* = 3)	Survival Rate (%)	Improved Survival Rate (%)
**Normal** *^a^*	0.901 ±0.05			**Normal** *^b^*	1.565 ± 0.04		
**Control** *^c^*	0.555 ± 0.02 ***	61.70		**Control** *^d^*	1.071 ± 0.04 ***	68.40	
**Donepezil** *^e^*	0.559 ± 0.03	62.10	0.66	**Donepezil** *^e^*	1.107 ± 0.04	70.70	3.33
**PHPB** *^e^*	0.569 ± 0.02	63.20	2.46	**PHPB** *^e^*	1.257 ± 0.09 ^#^	80.30	17.34
**1**	0.614 ± 0.01 ^#^	68.20	10.62	**1**	0.850 ± 0.07 ^##^	54.30	−20.67
**2**	0.577 ± 0.04	64.00	3.84	**2**	1.202 ± 0.07	76.80	12.23
**3**	0.567 ± 0.00	62.90	2.04	**3**	1.091 ± 0.18	69.70	1.84
**4**	0.729 ± 0.03 ^##^	80.90	31.27	**4**	0.904 ± 0.19	57.80	−15.56
**5**	0.528 ± 0.02	58.60	−4.92	**5**	1.109 ± 0.05	70.90	3.58

*^a^* DMEM (Dulbecco’s modified Eagle’s medium) without l-glutamic acid was added. *^b^*
l-DMEM (low-sugar Dulbecco’s modified Eagle’s medium) without sodium hyposulfite was added. *^c^*
l-glutamic acid was added as the damage agent. *^d^* Sodium hyposulfite was added as the damage agent. *^e^* Donepezil and PHPB were added as the positive controls. *** *p* < 0.001 vs. Normal; ^#^
*p* < 0.05 versus Control, ^##^
*p* < 0.01 versus Control.

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
