# Peer review of "Polycyclic Polyprenylated Acylphloroglucinol Derivatives from Hypericum acmosepalum"

_molecules, 2018, doi:10.3390/molecules24010050_

Round 1

Reviewer 1 Report

For the further study, I think need to more study for analyzing neuroprotective effect of them.

Author Response

Dear reviewer:

     Thank you for your comments on our manuscript. Your comments are all valuable and very helpful for improving our paper, as well as the important guiding significance to our further research. 

       Based on the suggestions from you, we had added our idea and expectation in future works in lines 227-229: “but further investigations of those compounds on neuroprotection and other activities are warranted. And via elucidating how molecules interact, this work will provide the necessary data to explore the relationship between neuroprotective activity and molecular structure.”

       Once again, thank you very much for your comments.

       Best regards,

       Yours sincerely,

       Jiao Wang

Reviewer 2 Report

In this paper, the authors extract 5 compounds from a dwarf shrub that is found in China. However, although the authors claim in the abstract (lines 20-22) that they characterise them by NMR spectroscopy, they only present the data and discussion from 2 of those. The names of those other 3 compounds were presented in lines 163, 175, and 178, but it is impossible for the reader or the reviewer to confirm or deny these assignments. Thus, the authors should either delete all references to these compounds, or, present the data.

Further comments:

Line 42-45: "Five..." These statements belong into the RESULTS section, not introduction
Lines 55-56: "degrees of unsaturation was required ten"... What does that mean?
Line 96: Figure 3. How was the ECD calculated? What are the experimental conditions such as solvent that were used to generate this figure?
Lines 101-102: "degrees of unsaturation was required ten"... What does that mean?
Lines 122-127. In this discussion of table 2, compounds donepezil and PHPB were presented, but not discussed. What is the compound "model" in table 2? On how many replicates were the standard deviations based on? That doesn't seem clear from the description in lines 190-203.
Lines 179-188: Further detail on the NMR experiments needs to be provided. What kind of experiments were done, under what experimental conditions?
Line 190: "MTT method" needs reference
Lines 196: The chemical MTT needs its abbreviation defined. Also, where do all the chemicals come from, which company, source?

Supplementary material:
1. Figure S6 seems to be the same as Figure 4 in the main text. Why is it duplicated?

Author Response

Dear reviewer:

       Thanks for your warm work earnestly, we had studied the comments carefully and had made correction. We are very sorry for our negligence of details, incorrect writing and inappropriate expressions. We hope that the correction will meet with approval. The main corrections in the manuscript and the responds to your comments are as the Word file.

       In all, your comments are quite helpful for improving our manuscript, and we had revised our manuscript point-by-point. Thank you again for your comments and suggestions! We hope this revision can make our manuscript more acceptable.

       Best regards,

       Yours sincerely,

       Jiao Wang

Reviewer 3 Report

This manuscript set out to further elucidate the PPAPs of Hypericum acmosepalum.  They did a remarkably fine job of isolating several compounds and elucidating the structures of two new PPAPs.

I find only one small issue with this manuscript.  Typically, the Introduction to a paper tells the reader the background for the research and instructs the reader in the purpose and intent of the research.  The results of the research are reported later in the Results section and Discussion section.  However, the authors of this paper included in the Introduction a partial report of the results.  Starting from the sentence in line 40 that begins, “A preliminary study…” to the end of the section in line 45 results from the study were reported.  These should be reserved for the Results section of the paper.

If the authors wish to expand further on the Introduction, as a reader I would welcome a brief review of some of the previous work done in their lab on PPAPs of Hypericum acmosepalum.

Beyond this, I found the paper to be very well researched and well documented.  The English is extremely good with only a awkward phrases in only a few places.

Author Response

Dear reviewer:

       We quite appreciate your insightful comments. Now we had revised our manuscript exactly according to your comments. The statement in lines 42-45: “Five...” (original manuscript) was deleted. At the same time, a brief word about the previous work done in our lab on PPAPs of Hypericum genus had been added as shown in lines 39-41 (“As a part of an ongoing research program aimed at the isolation, structural characterization and pharmacological evaluation of bioactive PPAPs from Hypericum genus [6, 19-20]”). 

       Moreover, some awkward phrases (lines 89, 55-56, 102-103) had been revised in manuscript.

       Once again, thanks very much for your comments and suggestions. We hope this revision can make our manuscript more acceptable.

       Best regards,

       Yours sincerely,

       Jiao Wang

Round 2

Reviewer 2 Report

The effort by the authors in revising the original manuscript is appreciated and the manuscript improved a lot. The authors may still want to address the following:

Lines 127-140 and 211-230. Table 2. Although it is written in the text that donepezil and PHPB are positive controls, it is still not clear what compounds "Normal" and "Control" are. There are separate entries for donepezil and PHPB in the table, and "Normal" and "Control". This may be obvious to the authors, but may not be obvious to the reader.

Author Response

Dear Reviewer:

       Thank you for your affirmation of the revisions and your new suggestions on our manuscript. Your suggestions are all valuable and very helpful for improving our paper. Based on your suggestions, we had finished the further revision of our manuscript, the details as follows:

       The description “L-glutamic acid and sodium hyposulfite were used as the damage agents” had been added in lines 124-125.

       In the Table 2, in order to make the Table 2 cleaner, the names of compounds “hyperacmosin A”, “hyperacmosin B”, “oxepahyperforin”, “sampsonione O” and “sampsonione L” had been revised to “1-5”. A new column (“Compounds”) was added into it. The notes about “Normal”, “Control”, “Donepezil” and “PHPB” had been provided (lines 135-139). Moreover, line 140: the typeface of word “Control” had been revised.

       Lines 216-224: the description “In the L-Glu, the DMEM (Dulbecco's modified Eagle's medium) without L-Glu was added to the cells of the normal group, the DMEM containing L-Glu (the final concentration of 27 mM) was added to the cells of other groups” and “In the OGD, the L-DMEM (Low-sugar Dulbecco's modified Eagle's medium) without sodium hyposulfite was added to the cells of the normal group, the L-DMEM containing sodium hyposulfite (the final concentration of 3.5 mM) was added to the cells of other groups” had been revised.

       Thank you again for your insightful suggestions. We hope this further revision can make our manuscript more acceptable. 

       Best regards,

       Yours sincerely,

       Jiao Wang